# Cleaning Efficacy of the XP-Endo® Finisher Instrument Compared to Other Irrigation Activation Procedures: A Systematic Review

**Dorina Lauritano** [1,*,†] **, Giulia Moreo** [1,†]**, Francesco Carinci** [2]**, Fedora Della Vella** [3] **,**
**Federica Di Spirito** [4]**, Ludovico Sbordone** [4] **and Massimo Petruzzi** [3]

1   Department of Medicine and Surgery, Centre of Neuroscience of Milan, University of Milano-Bicocca, 20126 Milan, Italy; moreo.giulia@gmail.com
2   Department of Morphology, Surgery and Experimental medicine, University of Ferrara, 44121 Ferrara, Italy; crc@unife.it
3   Interdisciplinary Department of Medicine, University of Bari, 70121 Bari, Italy; dellavellaf@gmail.com (F.D.V.); massimo.petruzzi@uniba.it (M.P.)
4   Department of Medicine, Surgery and Dentistry "Schola Medica Salernitana", University of Salerno, Via S. Allende, 84081 Baronissi, Italy; dispiritof@yahoo.it (F.D.S.); lsbordone@unisa.it (L.S.)
*   Correspondence: dorina.lauritano@unimib.it
†   These authors contribute equally to this work.

**Abstract:** Background. One of the most important aims of an endodontic treatment is to obtain the complete removal or reduction of root canal remaining filling material: Smear layer, bacteria, intra-canal medicaments. To meet this requirement, several irrigation activation techniques have been proposed. Our systematic review examined studies which analyzed the XP-endo Finisher (XPF) instrument efficacy in removing root canal debris during initial endodontic treatment or retreatment, comparing it with the efficacy of other irrigation activation protocols, such as passive ultrasonic irrigation (PUI), laser activation procedure (Er:YAG), and Self-Adjusting File system (SAF). Methods. A systematic review was conducted using PubMed, Chocrane Library, and Scopus databases, identifying 51 items. Thirty-four articles were excluded based on title, abstract, full text, and language. Seventeen randomized controlled trials were selected and consequently submitted to quality assessment and data collection. Results. Conventional needle irrigation (CNI) is the less effective irrigation technique, but it is still unclear whether XPF is able to guarantee greater debris removal than the PUI technique. Er:YAG laser has been proven to be more effective in apical third than XPF instrument. Conclusions. Further investigations are needed in order to establish which final irrigation activation procedure could reach the maximum root canal debris reduction.

**Keywords:** XP-endo Finisher; XP-endo Finisher effectiveness; irrigation activation protocols

## 1. Introduction

A successful endodontic procedure requires a proper shaping and irrigation. The linear movement and rotation of mechanical instruments in the root system produce the smear layer, a crystalline structure with 1–2 μ thickness, whose components (pulp residues, dentinal debris, bacteria, and their products [1]) can be found on the canal walls, root canal branches, and pressed into the dentinal tubules [2,3]. Moreover, during a canal retreatment, calcium hydroxide is used as intra-canal medicaments, in order to achieve the decrease of the bacterial amount. However, the remnants of this medicament, which have to be completely removed before permanent root canal filling, could obstacle the penetration of sealers and disinfectant, compromising the successful result of the endodontic therapy [4]. In addition,

the existence of different canal systems and anatomical variation (e.g., c-shaped canals, oval-shaped canals) complicates the treatment procedure. If residual intra-radicular infection occurs, a retreatment is required in order to eliminate the filling material and to reduce the presence of microorganisms. However, these goals are not always simple to achieve and may require adjunctive steps [5,6]. The only way to eliminate all of this type of debris is irrigation, a procedure that guarantees a positive impact on those areas of the canal system that cannot be reached by mechanical instruments [7]. One of the most common and profitable irrigation protocols involves the use of sodium hypochlorite (NaOCl), which solves necrotic tissues and reduces the bacterial load [8]. To fulfil its function, the irrigation procedure should provide the dispersion of the solution also within the inaccessible areas of the prepared canal walls, and, for this reason, several manual and mechanical irrigant agitation procedures have been proposed. In fact, a recent study by Conde et al. [9] demonstrated that irrigant activation improves tissue dissolution, overcoming the limitation of the conventional needle irrigation procedure (apical positive pressure), which is unable to clean the most apical portion of the root canal system. Examples of irrigant activation procedures are represented by the passive ultrasonic irrigation, the Self-Adjusting File system, or by the TRUShape 3D Conforming File. The passive ultrasonic irrigation technique activates the irrigant through ultrasonically oscillating small files or smooth noncutting wires, respecting the canal preparation [10]. The Self-Adjusting File system involves the use of hollow and thin cylindrical nickel–titanium instruments, which adapt to the cross-section of the root canal and operate with a constant flow of irrigant that is continuously replaced during the procedure [11]. Finally, the recent TRUShape 3D Conforming File is an S-shaped curve and blue color instrument, which preserves more tooth structure than ordinary NiTi instruments [12]. Recently a new mechanical cleaning system has been introduced: XP-Endo® Finisher (FKG Dentaire. La-Chaux-de-Fonds, Switzerland), that, with its revolutionary design, may enhance the ability to clean root canals with particular anatomical characteristics, reaching areas previously impossible to treat, but always preserving dentine. XP-Endo® Finisher is available in two sizes, ISO 025 and ISO30 (with 0% taper), and it can be used for retreatment cases (Finisher R, Reinforced) and for initial treatment (Finisher, a smaller version). It is made of nickel–titanium (NiTi) MaxWire alloy and, thanks to its small core size and its zero taper, it guarantees flexibility and excellent resistance to cyclic fatigue. Exploiting the shape-memory principles of NiTi alloy, this instrument is able to pass from the martensite-phase (straight shape, room temperature) to the austenite-phase (spoon-like shape) when exposed to body temperature, adapting to the specific morphology of the root canal.

### 1.1. Objectives

The objective of this systematic review was to analyze studies that tested the XP-Endo Finisher instrument efficacy in removing root canal detritus (e.g., smear layer, bacteria, intra-canal material remnants) during initial endodontic treatment or retreatment, comparing it with the efficacy of other irrigation activation protocols.

### 1.2. Clinical Question (PICO)

- P: XP-Endo® Finisher (FKG Dentaire, La-Chaux-de-Fonds, Switzerland);
- I: Efficacy of XP-Endo® Finisher in removing root canal debris during initial endodontic treatment or retreatment;
- C: Comparison between the cleaning efficacy of XP-Endo® Finisher and other final irrigation activation techniques;
- O: Evaluation of the efficacy of XP-Endo® Finisher as a new mechanical cleaning system compared to other final irrigation activation protocols.

## 2. Materials and Methods

### *2.1. Protocol and Registration*

The PRISMA statement [13] was used to select methods and inclusion criteria, since it provides a reliable protocol for systematic reviews.

### *2.2. Eligibility Criteria*

#### 2.2.1. Inclusion and Exclusion Criteria

All of the studies concerning the effectiveness of the new mechanical cleaning system XP-Endo® Finisher that met the following criteria were included in our systematic review:

Comparison between the efficacy of XP-Endo® Finisher in removing root canal debris and those of other irrigation agitation protocols;
Irrigation solution composed by NaOCl or EDTA, or both;
Quantitative data about the effectiveness of XP-Endo® Finisher; and
Randomized controlled trial.

We excluded case reports and other systematic reviews about this topic. Studies without quantitative data available were not included in our study.

#### 2.2.2. Search

We conducted a systematic literature search using the databases of PubMed, Cochrane Library, and Scopus in order to identify the most recent studies about XP-Endo Finisher® effectiveness, published by August 2019. Only articles written in English language were included, but no restrictions with regard to measurement methodology were imposed. As keywords, combined with the Boolean term "AND", "XP-Endo Finisher", "XP-Endo Finisher effectiveness", and "irrigation activation protocols" were used. The research was completed in October 2019.

#### 2.2.3. Study Selection and Data Collection Process

Two researchers (G.M. and D.L.) independently analyzed the title, abstract, and full text of all of the found items and, according to inclusion and exclusion criteria, they selected those that were eligible for this systematic review. Data collection was conducted by two reviewers (G.M. and D.L.), who extracted from each article the following information: Study design (randomized controlled trial), remaining root canal filling material that was measured after the irrigation protocols (smear layer, bacteria, $Ca(OH)_2$, triple or double antibiotic paste), and the different irrigation protocols that were used in order to remove the remaining material (XP-Endo Finisher, conventional needle irrigation, passive ultrasonic irrigation, Self-Adjusting File, TRUShape 3D, Er:YAG laser, CanalBrush, EndoActivator). Only irrigation protocols that used NaOCl or EDTA solutions were considered. Percentages and the scoring system by Lee et al. and van der Sluis et al. were used for the principal outcome measures. The flow chart that was used for this study is shown in Figure 1.

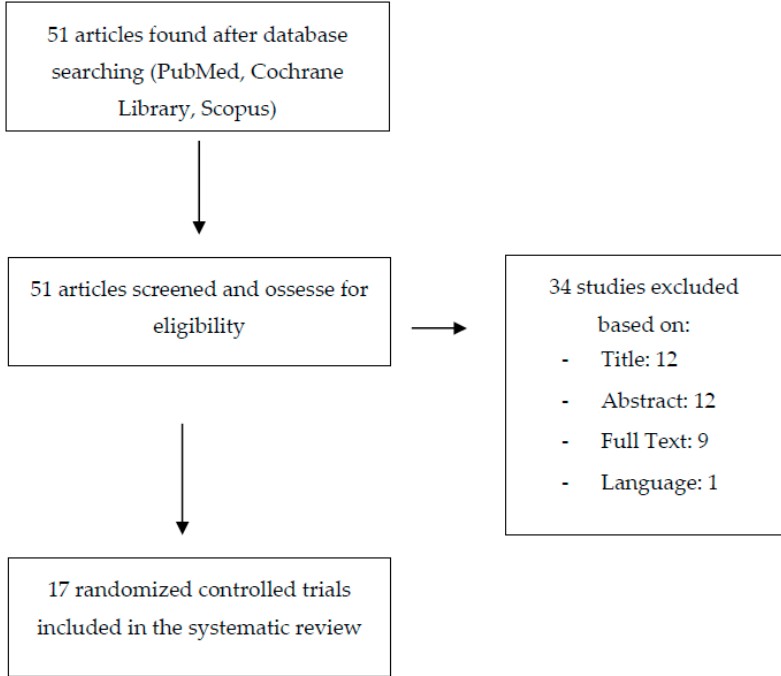

**Figure 1.** Flow chart of the publication assessment.

### 2.2.4. Quality Assessment

The quality level of the selected articles, evaluated with the Newcastle–Ottawa scale (NOS) [14], was considered high, since the lowest score was 6 and the highest one was 7 (Table 1). Most of the teeth samples analyzed in the articles were selected following reliable methodologies (radiographs, micro-TC, dental microscope) and the primary outcomes were recorded with adequate methods, such as confocal laser scanning microscopy, 3D Slicer, scanning electron microscope, and stereomicroscope. The evaluated quality parameters are shown in Supplementary Materials.

**Table 1.** Quality assessment of the included studies.

| Studies | Definition of Cases | Representativeness of Cases | Selection of Controls | Definition of Controls | Comparability | Exposure | Total |
|---|---|---|---|---|---|---|---|
| Alves et al., 2016 | + | + | - | + | +- | ++- | 6 |
| Azim et al., 2016 | + | + | - | + | +- | ++- | 6 |
| Bao et al., 2017 | + | + | + | + | +- | ++- | 7 |
| De-Deus et al., 2019 | + | - | + | + | +- | ++- | 6 |
| De-Deus et al., 2019 | + | - | + | + | +- | ++- | 6 |
| Elnaghy et al., 2017 | + | + | + | + | +- | ++- | 7 |
| Gokturk et al., 2016 | + | + | + | + | +- | ++- | 7 |
| Gokturk et al., 2016 | + | + | + | + | +- | ++- | 7 |
| Hamdam et al., 2017 | + | + | + | + | +- | ++- | 7 |
| Keskin et al., 2017 | + | + | + | + | +- | ++- | 7 |
| Kfir et al., 2017 | + | + | + | + | +- | ++- | 7 |
| Leoni et al., 2016 | + | + | + | + | +- | ++- | 7 |
| Turkaydin et al., 2017 | + | - | + | + | +- | ++- | 6 |
| Ulusoy et al., 2018 | + | + | + | + | +- | ++- | 7 |
| Uygun et al., 2016 | + | - | + | + | +- | ++- | 6 |
| Wigler et al., 2016 | + | + | + | + | +- | ++- | 7 |
| Zhao et al., 2019 | + | + | + | + | +- | ++- | 7 |

+: star assigned; -: star not assigned.

## 3. Results

### 3.1. Study Selection and Characteristics

The research through the PubMed, Cochrane Library, and Scopus databases identified a total of 51 citations and, after examining titles, abstracts, and full texts, only 17 articles were eligible for inclusion in this paper. Twelve articles were excluded based on the title, twelve based on the abstract, nine after reading the full text, and one because it was written in Japanese. The included studies were consequently submitted to quality assessment (using the Newcastle–Ottawa Scale) and data extraction.

This research selected 17 randomized controlled trials, whose characteristics are presented in Table 2, with reference to author and year of publication, study design, remaining root filling material measured, and irrigation protocols compared with XP-Endo Finisher. All of the articles were written in English language.

**Table 2.** List of included studies.

| Study | Design | Remaining Root Filling Material Measured | Irrigation Protocols Compared with XP-Endo Finisher |
|---|---|---|---|
| Alves et al., 2016 | RCT | Bacteria in mandibular molars | *2.5% NaOCl agitation*:<br>- PUI<br>- XPF |
| Azim et al., 2016 | RCT | Bacteria colonized in dentinal tubules | *17% EDTA + 6% NaOCl agitation with*:<br>- CNI (no activation)<br>- EndoActivator<br>- Er:YAG<br>- XPF |
| Bao et al., 2017 | RCT | Biofilm in apical root of single rooted premolars | *Three-step irrigation*:<br>- CNI (no activation)<br>- PUI<br>- XPF |
| De-Deus et al., 2019 | RCT | AHTD in oval-shaped canals | *5.25% NaOCl agitation + final flush with 17% EDTA for 2 min and bi-distilled water:*<br>- PUI<br><br>*5.25% NaOCl agitation with*:<br>- XPF |
| De-Deus et al., 2019 | RCT | Root filling remnants from oval-shaped canals | *2.5% NaOCl (in the first cycle), 17% EDTA (in the second one) agitation:*<br>- PUI<br>- XPF—R<br><br>*+ final flush with bi-distilled water* |
| Elnaghy et al., 2017 | RCT | Smear layer in curved canals | - CNI<br><br>*17% EDTA agitation with*:<br>- No additional agitation<br>- Agitated with BT2 file<br>- EndoActivator<br>- XPF<br><br>*+ 2.5% NaOCl + sterile saline solution* |
| Gokturk et al., 2016 | RCT | Ca(OH)$_2$ in single-rooted teeth | *2.5% NaOCl agitation with*:<br>- Beveled needle (no activation)<br>- Double side-vented needle (no activation)<br>- CanalBrush<br>- XPF<br>- Sonic irrigation<br>- PUI<br>- Er:YAG laser |

**Table 2.** *Cont.*

| Study | Design | Remaining Root Filling Material Measured | Irrigation Protocols Compared with XP-Endo Finisher |
|---|---|---|---|
| Gokturk et al., 2016 | RCT | DAP in single-rooted teeth | *2.5% NaOCl agitation with*:<br>- Beveled needle (no activation)<br>- Double side-vented needle (no activation)<br>- CanalBrush<br>- XPF<br>- Vibringe<br>- PUI<br>- Er:Yag laser |
| Hamdam et al., 2017 | RCT | Ca(OH)$_2$ in single-rooted teeth | *2.5% NaOCl agitation with*:<br>- PUI<br>- XPF |
| Keskin et al., 2017 | RCT | Ca(OH)$_2$ in single-rooted teeth | *5.25% NaOCl and 17% EDTA*:<br>- PUI<br>- EndoActivator<br>- CanalBrush<br>- XPF |
| Kfir et al., 2017 | RCT | Ca(OH)$_2$ in single oval canals | *4% NaOCl agitation with*:<br>- SAF<br>- PUI<br>- XPF<br>- CNI |
| Leoni et al., 2016 | RCT | AHTD in mesial root of mandibular molars | *2.5% NaOCl agitation with*:<br>- CNI (no activation)<br>- PUI<br>- SAF<br>- XPF |
| Turkaydin et al., 2017 | RCT | TAP in single-rooted teeth | - PUI<br>- CNI<br>- XPF |
| Ulusoy et al., 2018 | RCT | Organic tissue in straight root canals | - 2.5% NaOCl no activation<br>*2.5% NaOCl agitation with*:<br>- PUI<br>- XPF |
| Uygun et al., 2016 | RCT | Ca(OH)$_2$ in mandibular premolars | - 17% EDTA with needle irrigation (CNI), no activation<br>*17% EDTA agitation with*:<br>- RUShape 3D<br>- PUI<br>- XPF |
| Wigler et al., 2016 | RCT | Ca(OH)$_2$ in apical third of oval root canals | - 4% NaOCl with needle irrigation (CNI), no activation<br>*4% NaOCl activation with*:<br>- PUI<br>- XPF |
| Zhao et al., 2019 | RCT | AHTD in c-shaped canals | - 2% NaOCl + 17% EDTA with CNI, no activation<br>*2% NaOCl agitation + 17% EDTA with syringe and needle irrigation*:<br>- PUI<br>- XPF |

AHTD: Accumulated hard-tissue debris; Ca(OH)$_2$: Calcium hydroxide; CNI: Conventional needle irrigation; DAP: Double antibiotic paste; SAF: Self-Adjusting File; PUI: Passive ultrasonic irrigation; TAP: Triple antibiotic paste; XPF: XP-Endo Finisher; RCT: Randomized controlled trial.

The outcome measures reported in this systematic review refer to a total sample of 828 teeth, of which 245 teeth were treated with XP-Endo Finisher, 230 with passive ultrasonic irrigation, 184 with convention needle irrigation, 45 with Er:YAG laser, 40 with EndoActivator, 40 with CanalBrush, 16 with TRUShape 3D, and 28 with Self-Adjusting File.

### 3.2. Results of Individual Studies

According to six of the included studies, the XP-Endo Finisher instrument (XPF) showed a greater root canal cleaning efficacy than the passive ultrasonic irrigation technique (PUI) [5,15–19]. On the contrary, seven articles stated that the passive ultrasonic irrigation technique and XP-Endo Finisher showed no significant differences among them in removing root filling material from canal root, and that both protocols have a greater efficacy than conventional needle irrigation procedure (CNI) [10,12,20–24]. Gokturk et al. demonstrated in his studies [4,25] that PUI and laser-activated irrigation (LAI) were more efficient than XPF and CanalBrush (CB). All of the included researches recorded a lower cleaning performance of the standard needle procedure than the other irrigation methods (Table 3). The root filling material mean reduction (bacteria, AHTD, and organic tissues) obtained with XPF was equal to 63.84% [5,10,19,20,23,26]. Conventional needle irrigation and PUI guaranteed a mean reduction of 44.82% [10,19,23] and 64.12% [5,10,19,20,23], respectively. Azim et al. [26] stated that the mean bacterial reduction in apical third (at 150 μm) was 70% using Er:YAG laser, 47% using EndoActivator, and 23% using XP-Endo Finisher, demonstrating that the irrigation protocol with the aid of erbium:yttrium–aluminium–garnet (Er:YAG) laser had the greatest disinfection efficacy at higher depths. However, the same study recorded a greater efficacy of XP-Endo Finisher in killing apical third bacteria at 50 μm depth than the EndoActivator instrument and Er:YAG laser. Five of the included studies [4,12,22,24,25] used the scoring system by Lee et al. and van der Sluis et al. [27,28] in order to determine the amount of remaining debris in canal root systems (Figure 2). Out of 168 teeth that were irrigated with the aid of XPF (84 teeth) and PUI (84 teeth), 52.3% and 51.1% received a score of 2 (44/84 and 43/84), and 23.8% and 10.7% received a score of 3 (20/84 and 9/84), respectively. Empty grooves (score 0) were found in 12 teeth (14.2%) and in 15 teeth (17.8%), while in 8 teeth (9.5%) and 17 teeth (20.2%), respectively, calcium hydroxide was present in less than half of the grooves (score 1). Most of the teeth (39 out of 54) that were treated with CNI presented grooves still completely filled with calcium hydroxide (Score 3), and no teeth had empty grooves (Score 0) [12,22,24]. According to Kfir et al. [24], Uygun et al. [12], and Wigler et al. [22], XPF and PUI may ensure better results than CNI, although there are no significant differences among them. Gokturk et al. [4,25] claimed in their articles that no significant differences were found between PUI and laser-activated irrigation (LAI), and that these procedures eliminated root canal debris better than XPF, CanalBrush (CB), Vibringe, and CNI. Moreover, Gokturk et al. demonstrated that the XPF system showed similar results to those of CNI. One of the included articles [29] used a five-score scale to evaluate the remaining root filling material, highlighting that XPF and EndoActivator (EA) methods were more efficient than CNI, and that there were no significant differences between XPF and EA. In conclusion, almost all of the studies stated that none of the tested procedures were able to completely remove all root canal debris from all three root parts.

**Table 3.** Results of individual studies.

| Study | Filling Material Reduction | CNI | PUI | TS | CB | EA | Er:YAG | XPF | SAF |
|---|---|---|---|---|---|---|---|---|---|
| Azim et al., 2016 | Bacteria in apical third at 150 μm depth | 24% | | | | 47% | 70% | 24% | |
| De-Deus et al., 2019 | AHDT | | 62.66% | | | | | 62.67% | |
| De-Deus et al., 2019 | Root filling remnants in apical third | | 34.26% | | | | | 58.58% | |
| Elnaghy et al., 2017 | Smear layer in apical third | Total covering with a thick smear layer | | | | Thin smear layer and dentinal tubules partially open | | Thin smear layer and dentinal tubules partially open | |
| Gokturk et al., 2016 | Ca(OH)$_2$ in apical third (each protocol included 15 teeth) | | *Score 1–2*: 7 teeth *Score 0*: 1 tooth | | *Score 2–3*: 7 teeth *Score 1*: 1 tooth | | *Score 0*: 2 teeth *Score 1*: 5 teeth *Score 2*: 8 teeth | *Score 2–3*: 7 and 8 teeth | |
| Gokturk et al., 2016 | DAP (each protocol included 15 teeth) | | *Score 1*: 3 teeth *Score 2*: 9 teeth *Score 3*: 3 teeth | | *Score 1*: 6 teeth *Score 2*: 4 teeth *Score 3*: 5 teeth | | *Score 1*: 9 teeth *Score 2*: 5 teeth *Score 3*: 1 tooth | *Score 1*: 1 tooth *Score 2*: 9 teeth *Score 3*: 5 teeth | |
| Hamdam et al., 2017 | Residual Ca(OH)$_2$ mean values | | 3.6% | | | | | 2.1% | |
| Kfir et al., 2017 | Ca(OH)$_2$ (each protocol included 18 teeth) | *Score 3*: 16 teeth *Score 2*: 2 teeth | *Score 1*: 2 teeth *Score 2*: 13 teeth *Score 3*: 3 teeth | | | | | *Score 1*: 2 teeth *Score 2*: 13 teeth *Score 3*: 3 teeth | *Score 1*: 3 teeth *Score 2*: 11 teeth *Score 3*: 4 teeth |
| Leoni et al., 2016 | AHTD | 45.7% | 94.1% | | | | | 89.7% | 41.3% |
| Ulusoy et al., 2018 | Organic tissue | 28.5% | 52.3% | | | | | 85.0% | |
| Uygun et al., 2016 | Ca(OH)$_2$ in apical third (each protocol included 15 teeth) | *Score 1*: 8 teeth *Score 2*: 4 teeth *Score 3*: 4 teeth | *Score 0*: 14 teeth *Score 1*: 2 teeth | *Score 0*: 10 teeth *Score 1*: 5 teeth *Score 2*: 1 tooth | | | | *Score 0*: 12 teeth *Score 1*: 3 teeth *Score 2*: 1 tooth | |
| Wigler et al., 2016 | Ca(OH)$_2$ (each protocol included 20 teeth) | *Score 2*: 1 tooth *Score 3*: 19 teeth | *Score 1*: 3 teeth *Score 2*: 14 teeth *Score 3*: 3 teeth | | | | | *Score 1*: 2 teeth *Score 2*: 14 teeth *Score 3*: 4 teeth | |
| Zhao et al., 2019 | AHTD After instrumentation using: XP-Endo Shaper (XPS) or Reciproc Blue (RB) | XPS: 57.1% RB: 43.4% | XPS: 77.3% RB: 64.2% | | | | | XPS: 63.1% RB: 68.4% | |

CB: CanalBrush; EA: EndoActivator; SAF: Self-Adjusting File; TS: TrueShape.

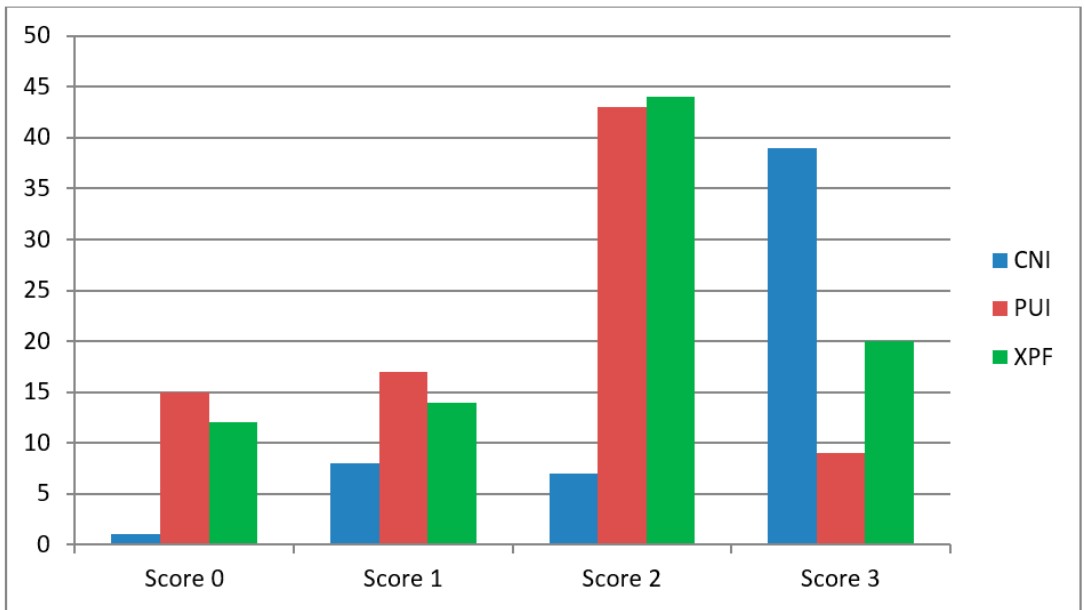

**Figure 2.** Distribution of scores for removing root canal debris in a sample of 222 teeth. Scoring system by Lee et al. and van der Sluis et al. → Score 0: Groove was empty; Score 1: Present in less than half of the grooves; Score 2: Present in more than half of the grooves; Score 3, The groove was completely covered [27,28].

## 4. Discussion

This systematic review had the objective of assessing the root canal cleaning effectiveness of the XP-Endo finisher instrument, both during initial endodontic treatment and retreatment, compared to the efficacy of other irrigant activation procedures. Studies included in this review tested the cleaning efficacy of different protocols using teeth with different anatomical morphologies: Straight or oval-shaped root canals, c-shaped canals, single-rooted teeth, or mandibular molars/premolars. The collected data suggested that CNI is the less effective irrigation technique, since it is not able to adapt to the particular anatomical characteristics of the root canal system and it does not allow to clean the root canal apical third [12,16,19,22–24,29]. Given its characteristics, the XP-Endo Finisher instrument surely provides a wider contact with canal walls, increasing cleaning efficacy, also in anatomically complex root canal systems, but it is still unclear whether it is able to guarantee greater debris removal than passive ultrasonic irrigation technique. PUI produces acoustic microwaves, which enhance the reaction of debris dissolution, generating hydrodynamic shear stress [19]. However, the ultrasonic tip could bind to the canal walls of curved canals, interfering with the acoustic streaming [20]. The laser activation procedure has been proven to be more effective in apical third than the XPF, EA, and CB methods [4,25,26,29], since it generates a shock wave effect, allowing a deeper penetration of irrigation solutions into dentinal tubules [26]. The SAF technique showed no significant differences from the XPF and PUI techniques in the study by Kfir et al. [24], but it has been demonstrated to be less effective in the study by Leoni et al. [10]. Our review highlights that further investigations are needed in order to establish which final irrigation activation procedure could reach the maximum root filling material reduction during endodontic treatment. Studies included in this paper also demonstrated that none of the above-mentioned methods ensure the total removal of root canal debris. Different types of root canals and different concentrations of irrigation solutions could be limitations of our study, leading to inhomogeneous results. Sodium hypochlorite (NaOCl) was used at 2, 2.5, 4, or 5.25% concentrations and ethylenediaminetetraacetic acid (EDTA) was used at 17% only. Moreover, root canal preparation of teeth samples in the included articles was performed by using different instruments, a condition that does not guarantee homogeneity. Three studies used the BT-Race rotary system [15,17,29]. Reciproc rotary files were used by De-Deus et al. [5,20], Gokturk et al. [4,25], and

by Zhao et al. [23]. Turkaydin et al. [18], Ulusoy et al. [19], and Uygun et al. [12] prepared the root canal system with the ProTaper Universal rotary system. Vortex Blue files, XP-Endo Shaper, the WaveOne Small File, and EndoSequence Endodontic File System were used only in one study, respectively [10,16,23,26]. Zhao et al. [23] tested the efficacy of the XP-Endo Finisher instrument compared with the passive ultrasonic irrigation technique and conventional needle irrigation. In all the three groups, significantly different hard-tissue debris reduction values were obtained depending on the type of the root canal preparation procedure. In the PUI group, the AHTD reduction values were 77.3 after the XP-Endo Shaper instrumentation and 64.2% after Reciproc rotary file preparation. In the XPF group, the values were 63.1% and 68.4%, respectively. In conclusion, additional studies should be conducted using uniform parameters with regard to teeth anatomical characteristics, concentrations of irrigation solutions, and root canal preparation systems.

## 5. Conclusions

Final irrigation activation procedure in endodontic treatment is needed in order to improve the root canal debris removal provided by the conventional needle irrigation technique. The laser activation procedure may have a greater efficacy than the XPF, EA, and CB methods, while the PUI protocol has been shown to be as effective as the XPF instrument or less effective than the latter. None of the analyzed protocols were able to completely remove root filling materials.

**Supplementary Materials:** The following are available online at http://www.mdpi.com/2076-3417/9/23/5001/s1.

**Author Contributions:** Conceptualization, D.L.; methodology, G.M.; validation, F.C., investigation, F.D.V.; data curation, F.D.S.; writing—original draft preparation, G.M.; writing—review and editing, D.L.; visualization, L.S.; supervision, M.P.

**Funding:** This research received no external funding.

**Conflicts of Interest:** The authors declare no conflicts of interest.

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
