# Peer review of "Cleaning Efficacy of the XP-Endo® Finisher Instrument Compared to Other Irrigation Activation Procedures: A Systematic Review"

_applsci, doi:10.3390/app9235001_

Round 1

Reviewer 1 Report

The present study aimed to perform a systematic review analyzed the XP-endo Finisher instrument efficacy in removing root canal debris during initial endodontic treatment or retreatment, comparing it with the efficacy of other irrigation activation protocols.

Although this is an important topic for the Endodontic literature, the present study has some issues and flaws that need to be corrected.

Line 28: What is CNI?

Line 58-66: Review this paragraph.

Line 107: Why did you not used other database? In the flow chart (fig 1) and line 135, the author also refer the Cochrane and Scopus database. Could you correct.

Line 274-276: Correct the reference.

Author Response

Milan 09/11/2019

Dear Reviewer,

Many thanks for the insightful comments and suggestions of the referees. We have made corresponding revisions according to their advice. Words in red are the changes we have made in the text. The language of the manuscript has also been extensively revised by a professional MDPI English language science editing service and all authors of this article have seen and approved the changes.

The revisions are as follows:

Line 28: What is CNI?

We have added, “conventional needle irrigation” beside the abbreviation “CNI”.

Line 58-66: Review this paragraph.

We have reviewed the paragraph. “Examples of irrigant activation procedures are represented by the passive ultrasonic irrigation, the Self-adjusting File system or by the TRUShape 3D Conforming File. Passive ultrasonic irrigation technique activates the irrigant through ultrasonically oscillating small files or smooth noncutting wires, respecting the canal preparation [10]. Self-adjusting File system involves the use of hollow and thin cylindrical nickel-titanium instruments, which adapt to the cross-section of the root canal and operate with a constant flow of irrigant, that is continuously replaced during the procedure [11]. Finally, the recent TRUShape 3D Conforming File is an S-shaped curve and blue color instrument, which preserves more tooth structure than ordinary NiTi-instruments [12].”

Line 107: Why did you not used other databases? In the flow chart (fig 1) and line 135, the author also refers to the Cochrane and Scopus database. Could you correct it?

àWe have used three databases, PubMed, Cochrane Library and Scopus. We found 37 articles on PubMed, 8 articles on Cochrane Library and 6 articles on Scopus.

We have corrected the sentence in line 123 à “We conducted a systematic literature search using the database of PubMed, Cochrane Library and Scopus”.

Line 274-276: Correct the reference.

We have corrected the reference: “ GA Wells, B Shea, D O’Connell, J Petersen, V Welch, M Losos, P Tugwell. The Newcastle-Ottawa Scale (NOS) for assessing the quality of nonrandomized studies in meta-analyses. Dep. Epidemiol. Community Med. Uni. 2014; http://www.ohri.ca/programs/clinical_epidemiology/oxford.asp”.

Thank you for receiving our manuscript and considering it for publication.

We appreciate your time and look forward to your response.

Yours sincerely,

Dorina Lauritano

Reviewer 2 Report

The authors provide a systematic review which analyzed the efficacy of XP-endo Finisher (XPF) instrument in removing root canal debris comparing to other irrigation activation protocols, such as passive ultrasonic irrigation (PUI), laser activation procedure (Er:YAG) and Self-adjusting file system (SAF). The manuscript is well written but for publication some minor revisions are necessary:

Abstract:

Li 28: Please indicate what is the meaning of CNI abbreviation

Introduction:

Li 37-41 " The linear movement and rotation of mechanical instruments in the root system  produce the smear layer, a crystalline structure with 1 – 2 μ thickness, whose components (that include organic and inorganic substances, such as pulp residues, dentinal debris, bacteria and their products [1] could be found on the canal walls, root canal branches and pressed into the dentinal tubules [2-3].

very long sentence, close bracket is missing, please rephrase this sentence.

Materials and Methods:

Li 123: The  brackets for literature 28,29 are missing.

Discussion:

Li 213-215: "Our review highlights that further investigations are needed, in order to establish which final irrigation activation procedure could reach the maximum root filling material reduction during endodontic treatment, becoming aware that none of the above-mentioned methods ensure the total remove of root canal debris. "

The sentence is also long please convert the informations to 2 or three short sentences.

The discussion part is generally very short, please add and discuss which studies and investigation are needed to answer the mentioned clinical questions. i

Please find attached in the pdf the result of plagiarism test please rewrite the suspicious sentences.

Author Response

Milan 09/11/2019

Dear Reviewer,

Many thanks for the insightful comments and suggestions of the referees. We have made corresponding revisions according to their advice. Words in red are the changes we have made in the text. The language of the manuscript has also been extensively revised by a professional MDPI English language science editing service and all authors of this article have seen and approved the changes.

The revisions are as follows:

Introduction

Li 28: Please indicate what is the meaning of CNI abbreviation

à We have added, “conventional needle irrigation” beside the abbreviation “CNI”.

Li 37-41 " The linear movement and rotation of mechanical instruments in the root system produce the smear layer, a crystalline structure with 1 – 2 μ thickness, whose components (that include organic and inorganic substances, such as pulp residues, dentinal debris, bacteria, and their products [1] could be found on the canal walls, root canal branches and pressed into the dentinal tubules [2-3]."

very long sentence, a close bracket is missing, please rephrase this sentence.

We have rephrased the sentence, making it shorter: “The linear movement and rotation of mechanical instruments in the root system produce the smear layer, a crystalline structure with 1 – 2 μ thickness, whose components (pulp residues, dentinal debris, bacteria and their products [1]) could be found on the canal walls, root canal branches and pressed into the dentinal tubules [2-3]”

Li 123: The brackets for literature 28,29 are missing.

We have removed the references since they are already mentioned later in the text.

Discussion

Li 213-215: "Our review highlights that further investigations are needed, in order to establish which final irrigation activation procedure could reach the maximum root filling material reduction during endodontic treatment, becoming aware that none of the above-mentioned methods ensure the total removal of root canal debris. "

The sentence is also long please convert the information to 2 or three short sentences.

We have rephrased the sentence, converting it in 2 sentences: “Our review highlights that further investigations are needed, in order to establish which final irrigation activation procedure could reach the maximum root filling material reduction during endodontic treatment. Studies included in this paper also demonstrated that none of the above-mentioned methods ensure the total removal of root canal debris”

The discussion part is generally very short, please add and discuss which studies and investigation are needed to answer the mentioned clinical questions.

We have added some sentences in the discussion part: “Different types of root canals and different concentrations of irrigation solutions could be limitations of our study, leading to inhomogeneous results. Sodium hypochlorite (NaOCl) was used at 2%, 2.5%, 4% or 5.25% concentrations and ethylenediaminetetraacetic acid (EDTA) was used at 17% only. Moreover, root canal preparation of teeth sample in the included articles was performed by using different instruments, a condition that doesn’t guarantee homogeneity. Three studies used the BT-Race rotary system [15-17-29]. Reciproc rotary files were used by De-Deus et al. [5-20], Gokturk et al. [4-25] and by Zhao et al. [23]. Turkaydin et al, [18], Ulusoy et al. [19] and Uygun et al. [12] prepared the root canal system with the ProTaper Universal rotary system. Vortex Blue files, XP-endo Shaper, the WaveOne Small file and EndoSequence Endodontic File System were used only in one study respectively [10-16-23-26]. Zhao et al. [23] tested the efficacy of XP-endo Finisher instrument compared with passive ultrasonic irrigation technique and conventional needle irrigation. In all the three groups, significantly different hard-tissues debris reduction values were obtained depending on the type of the root canal preparation procedure. In PUI group the AHTD reduction values were 77.3 after XP-endo Shaper instrumentation and 64.2% after Reciproc rotary files preparation. In XPF group the values were 63.1% and 68.4% respectively. In conclusion, additional studies should be conducted, using uniform parameters with regard to teeth anatomical characteristics, concentrations of irrigation solutions and root canal preparation system.”      

Please find attached in the pdf the result of plagiarism test please rewrite the suspicious sentences.

We have rewritten the suspicious sentences, following the pdf with plagiarism test results.    

Thank you for receiving our manuscript and considering it for publication.

We appreciate your time and look forward to your response.

Yours sincerely,

Dorina Lauritano

Reviewer 3 Report

Did you register with PRISMA (PROSPERO http://www.crd.york.ac.uk/PROSPERO) your systematic review?

Explain detailed the quality of studies in the Newcastle-Ottawa (NOS) (Wells, 2014)

Author Response

Milan 09/11/2019

Dear Reviewer,

Many thanks for the insightful comments and suggestions of the referees. We have made corresponding revisions according to their advice. Words in red are the changes we have made in the text. The language of the manuscript has also been extensively revised by a professional MDPI English language science editing service and all authors of this article have seen and approved the changes.

The revisions are as follows:

Explain detailed the quality of studies in the Newcastle-Ottawa (NOS) (Wells, 2014)

We have added Table 1 and Additional File 1, in order to explain the details of the quality assessment of the included studies.

Did you register with PRISMA (PROSPERO http://www.crd.york.ac.uk/PROSPERO) your systematic review?

Our systematic review is not registered with PRISMA (PROSPERO http://www.crd.york.ac.uk/PROSPERO).

Thank you for receiving our manuscript and considering it for publication.

We appreciate your time and look forward to your response.

Yours sincerely,

Dorina Lauritano

Round 2

Reviewer 1 Report

Dear author,

Congratulations for the new version of the paper.

Reviewer 2 Report

Thank you for the revision, all requested changes were done. Please note that page 3 is empty.

Reviewer 3 Report

The actual modifications in this version are correct. For the next systematic review I will encourage you to register the study in PROSPERO.